# Recent Advances in Pineoblastoma Research: Molecular Classification, Modelling and Targetable Vulnerabilities

**DOI:** 10.3390/cancers17050720

**Published:** 2025-02-20

**Authors:** Zhe Jiang, Michelle S. Allkanjari, Philip E. D. Chung, Hanna Tran, Ronak Ghanbari-Azarnier, Dong-Yu Wang, Daniel J. Lin, Jung Yeon Min, Yaacov Ben-David, Eldad Zacksenhaus

**Affiliations:** 1Toronto General Research Institute, University Health Network, 101 College Street, Max Bell Research Centre, Suite 5R406, Toronto, ON M5G 1L7, Canada; michelle.allkanjari@mail.utoronto.ca (M.S.A.); hanna.tran@mail.utoronto.ca (H.T.); ghronak@gmail.com (R.G.-A.); dongyu.wang@utoronto.ca (D.-Y.W.);; 2State Key Laboratory for Functions and Applications of Medicinal Plants, Guizhou Medical University, Guiyang 550014, China; 3Natural Products Research Center of Guizhou Province, Guiyang 550004, China; 4Department of Medicine, University of Toronto, Toronto, ON M5S 1A1, Canada

**Keywords:** pineoblastoma, brain cancer, RB1, MYC, Dicer1, Drosha, classification, metastasis, mouse models, precision medicine

## Abstract

Pineoblastoma is a difficult-to-treat pediatric brain cancer that has traditionally been viewed and treated as a single disease. However, over the past five years, pineoblastoma has been recognized as comprising four major subtypes driven by unique oncogenic alterations, affecting different age groups and leading to distinct clinical outcomes. Additionally, mouse models for these distinct pineoblastoma subtypes have been developed, enabling the assessment of targeted therapies. These advances will likely usher in a new era of translational and clinical research and improve outcomes for this lethal disease.

## 1. Introduction

Pineoblastoma (PB) is a rare, malignant tumor on the pineal gland, with an annual incidence of approximately 5 per 2,000,000 in the United States [1,2,3]. PB incidence is higher in Black compared to White patients, and in non-Hispanic compared to Hispanic patients, affecting infants, children, and occasionally adults. PB patients experience elevated intracranial pressure that causes headaches, nausea, vomiting and related symptoms [4]. The overall survival (OS) rate is approximately 15% in patients ≤5 years old [5]. Metastatic disease is incurable [6]. Due to the rarity of the disease, most available clinical data on PB are based on case reports and small studies with a limited number of patients [7]. Consequently, treatments for medulloblastoma and central nervous system primitive neuroectodermal tumors (CNS-PNETs) have historically served as the basis for PB care. Current standard treatment comprises maximal surgical resection, adjuvant radiation, and systemic chemotherapy, which, while somewhat effective, often results in major neurocognitive decline. Thus, there is an unmet need for larger-scale studies to elucidate the biology and possible heterogeneity of PB and for the generation of relevant mouse models to improve diagnosis, modeling, and treatment. This review summarizes recent advances in the molecular stratification of PB patients, which have uncovered four major subtypes, and the development of genetically engineered mouse models for these PB subtypes, enabling subtype-specific preclinical trials.

## 2. The Pineal Gland

The pineal gland is a small endocrine gland that regulates the circadian rhythm by secreting the hormone melatonin. This hormone controls the sleep–wake cycle, feeding, temperature, and heart rate [8,9]. The pineal gland secretes melatonin during the night; therefore, blood melatonin levels peak at night and decrease during the day [10]. The rhythmic secretion of melatonin is regulated by light in a 24 h endogenous cycle, which in humans is detected by the retina and transmitted to the pineal gland [11]. Melatonin is produced in the pineal gland in a two-step reaction: first, serotonin is converted into N-acetylserotonin by arylalkylamine N-acetyltransferase (AANAT), which is subsequently converted into melatonin by hydroxyindole-O-methyltransferase (HIOMT) [10].

The pineal gland develops from the dorsal encephalon above the area where the third ventricle forms between the two hemispheres of the brain. It evaginates from the dorsal encephalon to form a lobular-shaped vesicle. In humans, the pineal gland is located on the dorsal aspect of the brainstem, whereas in rodents, it is located on the dorsal surface of the brain between the cerebral cortex and the midbrain [12] (Figure 1). The pineal gland is composed of several cell types including pinealocytes, astrocytes, microglia, endothelial cells, neurons, and peptidergic neuron-like cells [13,14]. The pinealocyte can be further divided into α- and β-pinealocytes, which differ by the expression of acetylserotonin O-methyltransferase (Asmt) [14]. The pinealocyte is a club-shaped cell, which accounts for approximately 90–95% of cells in the pineal gland and expresses retinal antigens such as synaptophysin, S-antigen and opsin [15]. The astrocytes are star-shaped cells, smaller than the pinealocyte, which occasionally express the glial fibrillary acid protein (GFAP). Phagocytes are mostly present in the perivascular space.

The brain and the spinal cord are protected by blood–brain barrier (BBB). Microvasculature of the central nervous system (CNS) is composed of non-fenestrated vessels with tight regulation, allowing selective movement of ions, molecules and cells required for neuronal function [16]. While BBB protects the brain and spinal cord from toxins, inflammation and pathogens, it also blocks entrance of certain drugs into the CNS, thus limiting the delivery of therapeutics into the brain and spinal cord [16]. There is evidence, however, that drug delivery to the pineal gland is relatively unhindered by the BBB [13].

## 3. Tumors of the Pineal Gland

Pineal tumors are categorized into pineal parenchymal tumors (PPTs) and germ cell tumors (GCTs) [17,18,19,20]. PPTs include PB, the major subtype, as well as PPTs with intermediate differentiation, pineocytomas (benign lesions), and papillary tumors of the pineal region (PTPRs) [21,22,23,24]. Additionally, some rare pineal gland tumors arise from non-pinealocytes. These include atypical pleomorphic astrocytoma, glioblastoma, ganglioglioma, oligodendroglioma, meningioma, melanoma, and solitary fibrous tumors [25,26,27,28,29,30].

## 4. Pineoblastoma

PB is a rare tumor of the pineal gland affecting less than 1% of all pediatric brain cancer patients. It is the most malignant tumor of the pineal gland accounting for 30–40% of all PPTs [31,32,33]. Common symptoms of PB include nausea, vomiting, ocular disturbances, headaches, fatigue, problems with coordination; weakness on one side of the body; seizures; diminished growth rate; and abnormal pupils, all of which are related to hydrocephalus, the buildup of cerebrospinal fluid in the brain [34]. Risk factors for PB include germline mutations in two tumor suppressors, *RB1* and *DICER1*. PB incidence increases in children with bilateral retinoblastoma due to a germline mutation in the RB1 tumor suppressor, a condition referred to as ‘trilateral retinoblastoma’ [35,36,37]. Similarly, children with DICER1 syndrome, caused by a germline mutation in *DICER1*, are also predisposed to PB [38,39,40,41]. While germline mutations in *RB1* and *DICER1* predispose to PB, sporadic cases may involve somatic loss of *DROSHA*, duplication of *PDE4DIP* or *cMYC* gain [42,43,44,45].

In this context, the parallel between retinoblastoma and pineoblastoma is noteworthy. Both are driven by germline mutations and loss of heterozygosity (i.e., complete loss) of *RB1*. However, while a small group of retinoblastomas with intact, phosphorylatable pRB, is driven by *MYCN* amplifications [46], a major subset of pineoblastomas is induced by amplifications or stabilization of *cMYC* [43,44,45]. Notably, at least in mice, *cMYC* and *MYCN* are genetically interchangeable [47]. Thus, the reason for this tumor-specific preference for *cMYC* vs. *MYCN* amplification may be related to the effect of nearby genes that are co-amplified in each oncogenic event [48,49], or differences in mechanisms of stabilization and regulation the of respective onco-proteins [50].

The incidence of PB varies between males and females depending on the subtype (see below). It is most frequently found in infants and children, but occasionally occurs in adults [6,7,31,51,52]. The overall survival (OS) rate of patients with PB is approximately 54–60% [7,52], but this varies dramatically in specific subtypes. Of 299 patients analyzed in one study, children older than 5 years showed better OS (5-year survival rate = 57%) compared to children ≤5 years old (5-year survival rate = 15%) [7]. Moreover, OS was better for older patients treated with radiotherapy [52].

PBs are classified by the extent of tumor spread and metastasis as follows:

M0: Localized tumor with no sign of spread to other parts of the body.

M1: Tumor cells spread to the cerebrospinal fluid.

M2: Tumor cells spread to other parts of the brain.

M3: Tumor cells spread to the spine.

M4: Tumor cells spread outside the central nervous system.

PB is also classified based on histology and growth rate. In general, PBs are high-grade (WHO IV) [53] with proliferation index of 36.4% (as determined by staining for MIB-1, an antibody directed against different epitope of the proliferation-related antigen Ki67), which is significantly higher than lower grade tumors of the pineal gland [54,55]. Histologically, pineoblastomas are similar to medulloblastomas and primitive neuroectodermal tumors (PNETs) [21]. PB cells are densely packed with features of a high nuclear to cytoplasmic ratio, irregular and neomorphic nuclei, and scant cytoplasm [56]. Homer Wright rosettes and Flexner–Wintersteiner rosettes are often observed [21,57]. Immunophenotype of PB includes high levels of neuron-specific enolase (NSE), synaptophysin, neurofilament (NF) and chromogranin A but absence or very low expression of glial fibrillary acidic protein (GFAP) [56].

## 5. Metastatic Pineoblastoma

Metastasis is the major cause of death in certain malignancies such as breast cancer. It involves a metastatic cascade of events, including invasion, intravasation into the surrounding vasculature or lymphatic system, survival in the circulation, extravasation from the vasculature into secondary tissues and colonization at distal sites, all of which can be therapeutically targeted ([58,59] and references therein). In pineoblastoma, patients with metastases have worse outcomes compared to those without metastases [6], but complications from primary disease and therapy itself are a major cause of mortality and decline in quality of life. Metastatic PB characterize the RB1- and MYC- subtypes (see below). Metastases are often observed in the spinal cord and brain, but rare metastases also occur outside of central nervous system [34,60,61]. The dissemination of brain tumors was traditionally thought to occur passively via cerebrospinal fluid (CSF). However, a recent study has shown that for medulloblastoma, tumor cells can spread via the blood stream [62]; whether PB also spreads through this hematogenous route is yet to be determined.

The dissemination of diverse types of cancers is facilitated by interactions with the nervous system [63,64]. These interactions include electrochemical communication through neuron-cancer synapses, and paracrine signaling from neurons to cancer cells. Brain invasion of glioblastoma is potentiated by neuron-cancer-like synapses that induce Excitatory Postsynaptic Currents (EPSCs) through calcium-permeable AMPA (α-amino-3-hydroxy-5-methyl-4-isoxazole propionic acid) receptors located on glioma cells, leading to induction of cell proliferation and invasion [65,66]. In breast cancer, a paracrine signal from sensory neurons that are recruited to tumors induces cell death, and release of single-stranded RNA (ssRNA). The latter binds and induces Toll-like Receptor 7 (TLR7) on tumor cells, and a non-canonical TLR7 signaling that promotes dissemination [67]. Whether PB spreads via similar mechanisms involving synaptic connections or paracrine signaling remains an important area of research, as these processes are amenable to therapeutic interventions.

## 6. Current Treatment of Pineoblastoma and Ongoing Clinical Trials

Current treatment options for PB include maximal resection of tumor bulk, radiotherapy and chemotherapy with drugs such as cisplatin, vincristine, and cyclophosphamide [6,68]. Univariate analysis has established a correlation between children diagnosed at a young age (≤5 years) and an exceedingly poor prognosis [7]. This relationship is amplified if the patient has metastatic disease at diagnosis, rendering treatment of these groups particularly challenging. While gross total resection is ideal, it is clinically challenging due to the location of the pineal gland and its surrounding vascular structures [69]. In addition, children under the age of 3 are not treated with radiation as this can result in major neurocognitive decline [70]. Chemotherapy has been brought into question as studies suggest it is only marginally more effective than radiotherapy and surgery [7]. These observations underscore the lack of effective therapeutic strategies that minimize toxicity in PB patients, particularly in those aged ≤5 years old with metastatic disease. Thus, there is an urgent need to develop new therapies for PB that can improve patient outcomes while minimizing off-target cytotoxic effects.

Several ongoing clinical trials (https://clinicaltrials.gov/search?cond=Pineoblastoma, 20 December 2024) are listed in Table 1, with additional details, including links to recently completed trials, in Appendix A. Common themes from these trials are that (i) with one exception (NCT02596828; Appendix A), which involved PB patients, all other trials target various brain tumors including PB; and (ii) none of the trials stratified PB or targeted a specific subtype.

These clinical trials include an ongoing clinical trial (NCT02574728–recruiting) testing response to sirolimus (rapamycin), an immunosuppressive agent and inhibitor of the mammalian target of rapamycin (mTOR), which is a serine/threonine-specific protein kinase that regulates cell growth, proliferation, and survival; celecoxib, a COX-2 inhibitor and nonsteroidal anti-inflammatory drug; etoposide, a topoisomerase II inhibitor; and cyclophosphamide nitrogen mustard alkylating agent.

NCT02596828, a prospective pilot trial on children, adolescent and young adult PB patients with, irinotecan, a topoisomerase I inhibitor, rapamycin, an mTOR inhibitor; dasatinib, a protein tyrosine kinase inhibitor with high affinity to the ABL kinase domain; and temozolomide, which alkylates and methylates DNA, leading to cell death. Inhibition of topoisomerase 1 leads to abnormal R-loop accumulation and synthetic lethality with MYC-driven cancer [71], suggesting that the results from this trial may be deconvoluted based on MYC status. As discussed below, inhibition of mTOR via rapamycin may also target MYC-driven tumors.

Another trial involving irinotecan (NCT01217437) for various brain tumors including PB also administered bevacizumab, a vascular endothelial growth factor A (VEGF-A) inhibitor, and temozolomide. Results from this completed trial may also be analyzed based on MYC status.

The NCT00867178 trial tested combination of proton beam radiation therapy with cyclic therapy with vorinostat and isotretinoin, followed by vincristine; cisplatin; cyclophosphamide; and then etoposide phosphate for medulloblastoma, pineoblastoma and supratentorial Embryonal Tumor. Vorinostat is a histone deacetylase (HDAC) inhibitor, and as discussed below, may be effective against MYC-driven brain tumors, raising the possibility that clinical outcome from this trial, which has been completed, may be reassessed after patient selection based on MYC status in PB and other brain lesions.

The NCT02095132 trial on diverse central nervous system embryonal neoplasms used irinotecan hydrochloride together with adavosertib (AZD1775, MK-1775), a WEE1 inhibitor that, as described below, targets RB deficiency along the DNA-damage response pathway [72,73]. It would be important to test whether RB-deficient or MYC-induced PBs or other brain tumors in this cohort were significantly more susceptible to this drug regimen compared to other tumors with normal levels of RB and MYC.

A few additional ongoing trials of interest involve CAR-T cell therapy: NCT03638167, involving EGFR806-specific chimeric antigen receptor (CAR) T cells; NCT03500991 using HER2-specific chimeric antigen receptor (CAR) T cells, and SCRI-CARB7H3(s); B7H3-specific chimeric antigen receptor (CAR) T cells; and NCT06357377, a safety, dosing, and delivery trial for NEO100, a purified version of the natural monoterpene perillyl alcohol, which can enable delivery of BBB impermeable therapeutics, as well as NCT05064306, 131I-omburtamab, a radioimmunoconjugate consisting of the iodine 131-radiolabeled murine IgG1 monoclonal antibody 8H9 directed against the surface immunomodulatory glycoprotein 4Ig-B7-H3 (CD276) expressed on the cell membranes of diverse types of tumors of neuroectodermal, mesenchymal and epithelial origin.

In summary, current therapy and past or ongoing clinical trials treat all PBs as a single disease, potentially overlooking opportunities to detect subtype-specific responses. For instance, MYC-PBs and RB1-PBs are sensitive to specific inhibitors, and these patients likely respond differently to treatment regimens. As described below, recent stratification of PBs into unique groups with distinct oncogenic drivers and clinical outcome, and advances in the development of immune-competent mouse models for each subtype begin to uncover patient-tailored novel therapeutic approaches including immunotherapy for this lethal childhood disease.

## 7. Molecular Classification of Pineoblastoma

Due to the rarity and limited accessibility of PB samples, bioinformatics analyses of its genomic and epigenomic landscapes have been scarce. Limited gene expression and methylome data of PB samples were initially analyzed together with other brain tumors [74,75,76]. PB was clustered close to group 3 and group 4 medulloblastoma as well as to retinoblastoma. In these studies, PBs were subdivided into two large subgroups: PB-A, clustered together with retinoblastoma, consisted of patients who are diagnosed as “intracranial retinoblastoma” including children with *RB1* germline mutation; and PB-B, whose underlying biology was undetermined [75]. In the past 4–5 years, major oncogenic drivers and subtypes of PB have been identified.

As noted, some patients with familial retinoblastoma, which carry *RB1*-germline mutation, develop both retinoblastoma, an ocular tumor, and PB, a condition termed “trilateral retinoblastoma” [35,36,37]. Although advanced chemotherapy has significantly improved outcome for trilateral retinoblastoma patients, 5-year OS remains low at 44% [77]. Germline mutations in *DICER1*, which cleaves pre-miRNA to form mature miRNA, also predispose individuals to PB [78]. Such germline mutations, Dicer syndrome, increase the risk of tumors of the kidney, thyroid, ovary, cervix, testicle, brain, eye and lining of the lung without loss of heterozygosity (LOH) [38,41,79]. LOH of *DICER1*, leading to complete loss of this tumor suppressor, is associated with PB. In an early study, 4 *DICER1* mutations were observed in 18 PB patients [78]. In addition, homozygous deletion of *DROSHA*, a miRNA-processing enzyme upstream of *DICER1*, and duplication of *PDE4DIP*, phosphodiesterase 4D-interacting protein, have been identified in sporadic cases of PB [42].

Several reports have recently classified pineoblastomas at the molecular level using Genome-wide DNA methylation profiling, whole-genome sequencing (WGS), whole-exome sequencing (WES) and transcriptomic profiling [43,44,45]. The first such study from 2020 analyzed 195 tumors of the pineal region by WGS and identified 3 major molecular subtypes of PB with alterations in microRNA processing (*DICER1*, *DROSHA* or *DGCR8*), *MYC*, or *RB1* [43]. These subgroups differed in age of onset, sex, chromosomal alterations, and overall survival (OS). In a parallel study, 91 patients with PB or supratentorial primitive neuroectodermal tumor (sPNETs/CNS-PNETs) were analyzed and 5 groups were delineated [44]. Groups 1 and 2 exhibited homozygous loss or functional alterations in the miRNA biogenesis genes DICER1, DROSHA or DGCR8. Group 3 contained in-frame insertions in KBTBD4; group 4 exhibited *RB1* loss or miR-17/92 gain [80]; and group 5 included recurrent gain or amplification of the *cMYC* oncogene. While group 1–3 included older children (median ages 5.2–14.0 years) with intermediate to excellent survival (5-year OS of 68.0–100%), *RB1* and *MYC* PB patients were younger (median age 1.3–1.4 years) with dismal survival rates (5-year OS 37.5% and 28.6%, respectively).

A subsequent study in 2021 on 221 patients with PBs and pineal parenchymal tumors of intermediate differentiation (PPTIDs) corroborated these results with similar classification [45]. Specifically, PBs were molecularly stratified into four major clinical subtypes: PB-miRNA1, PB-miRNA2, PB-RB1 and PB-MYC/FOXR2 [45]. A fifth group comprised PPTIDs and exhibited alterations in KBTBD4, a substrate-specific adapter of a BCR (BTB-CUL3-RBX1) E3 ubiquitin ligase complex, representing group 3 in the study above [44].

In the 2021 study, miRNA1/2 PBs also include the oldest group of patients with a median age of 8.5 and 11.6 years, respectfully. The ratio of males to females is approximately 1:1.6 in miRNA1, and 1:0.63 in miRNA2. These groups are more prone to localized disease and are characterized by alterations of miRNA-processing genes. PB-miRNA1 tumors exhibit loss of *DICER1* (located on cytogenetic band 14q32.13), *DROSHA* (5p13.3), or *DGCR8* (22q11.21) [81]. While Drosha and Dicer1 are both RNA III endonucleases involved in the production of mature miRNA [82], DGCR8 acts as a microprocessor to guide Drosha during the cleavage of the pre-mRNA [83]. PB-miRNA2 lesions also harbor alterations in *DICER* and *DROSHA* but not *DGCR8* [45]. *DICER* alterations in PB-miRNA2 consistently co-occur with loss of chromosome 14q (chr14q), leading to a complete loss of function, which is not observed in the PB-miRNA1 subtype. Patient outcomes in the PB-miRNA1 subtype are relatively favorable, with 5-year progression-free survival (PFS) and OS rates of 56.7% and 70.3%, respectively. PB-miRNA2 patients had even better 5-year PFS (86.1%) and OS (100%).

Among young children less than three years of age, 40% were classified as PB-RB1 and 45% as PB-MYC/FOXR2, both exhibiting rapid progression and dismal outcome. 81% of sequenced PB-RB1 tumors show focal deletion, frameshifting/truncating variants (FTV), or combinations of such alterations in the tumor suppressor *RB1*. The chromosomal locations of these genes are: *RB1* (Chr 13q14.2); *MYC* (8q24.21); and *FOXR2* (Xp11.21). This group also exhibit a recurrent focal gain of *miR-17/92* (Chr 13q31.3), which acts to stimulate proliferation, angiogenesis and cell survival, while inhibiting cell differentiation [80]. As in retinoblastoma [84], frequent chr 16 loss (mostly chr16q alone or together with 16p) accompanied by gain of chr 1q and 6p are observed. PB-RB1 has the highest rate of metastasis, with 69% of patients exhibiting metastatic disease at the time of diagnosis. This is compared to 43% metastatic rate for PB-MYC/FOXR2, 42% for PB-miRNA1 and 16% for PB-miRNA2 patients. OS and PFS rates for PB-RB1 patients are extremely low at 23.8% and 29.8%, respectively.

Copy-number-variation (CNV) data in PB-MYC/FOXR2 tumors revealed frequent loss of chr 16q and large gain of chr 8q, which encompasses *cMYC* (8q24.21). Several cases exhibited focal *MYC* amplification. The remaining non-MYC-amplified tumors showed overexpression of the proto-oncogene *FOXR2* [45]. FOXR2 directly interacts with MYC to promote its transcriptional activity and tumorigenesis [85,86,87]. Frequent (6 of 8) intrachromosomal re-arrangements leading to induction of FOXR2 were also found in neuroblastoma; one tumor without such re-arrangement and FOXR2 induction exhibited focal MYC amplification [76]. Subsequent work showed that FOXR2 stabilizes MYCN protein in non-MYCN-amplified neuroblastoma patients [88], and it presumably also stabilizes non-cMYC-amplified cancers. PB-MYC/FOXR2 comprises the youngest cohort of patients with a median age of just 1.3 years old [45]. Like the PB-RB1 group, high rates of metastatic disease are observed in these patients and, accordingly, PB-MYC/FOXR2 patients have dismal 5-year OS (19.2%) and PFS (16.7%) rates [45]. Interestingly, while the male to female ratio is nearly 1:1 for RB1-PB patients, the MYC-PB subtype exhibits a high male to female ratio of over 3 fold.

Response to standard treatment (surgical resection, radiation, and/or standard/high-dose chemotherapy) differs in the four subtypes. While PB-miRNA1/2 lesions respond well, PB-RB1 and PB-MYC/FOXR2 are relatively resistant to treatment. The latter two groups include the youngest PB patients, less than 3 years of age, that cannot receive radiation therapy [5,7,89], and as noted, children with PB who do not receive radiation exhibit worse survival outcomes [5]. However, radiation in MB was shown to enhance recurrent disease and metastatic spread by selecting for rare variants with mutations in TP53 or the TP53 pathway [90]. Thus, conventional radiation is a double-edged sword that reduces tumor burden but selects for aggressive TP53-deficient variants. Whether advanced radiation approaches such as targeted alpha-particle therapy [91], image-guided pencil beam proton therapy [92] (NCT01063114) or other approaches such as iodine 131 conjugates discussed above (NCT05064306), can overcome these adverse effects in pediatric brain cancer is yet to be determined. Either way, RB-loss and MYC-gain PB patients experience exceedingly poor outcomes and should be prioritized for the development of novel and targeted therapies.

## 8. Xenograft Models of Pineoblastoma

Xenografts of human brain tumors into immunocompromised mice, implanted subcutaneously rather than intracranially simplifies the process but fails to recapitulate the distinctive tumor microenvironment of the pineal region or brain tissue with its unique cellular niche. Very few PB patient-derived xenografts (PDXs) from cells or tissue have so far been propagated by transplantation directly into the murine pineal region or adjacent compartments, thereby better reflecting the tumor’s native milieu, growth patterns, and invasive properties. In one study, PB was propagated as tumorspheres in culture and as PDX in immune-deficient mice [93]. However, even these human PB cells are extremely difficult to propagate in culture and have a limited replication span [94]. In contrast, we were readily able to propagate and interrogate mouse Rb/p53-deficient PB cells in culture [94], possibly because of p53 loss, which is required for the development of PB in mice. The establishment of immune-competent mouse models for each PB subtype may provide vital platforms for studying the biology of each subtype, testing and screening for novel therapeutics, and exploring the role of genetic and epigenetic factors in tumor initiation and progression.

## 9. Subtype-Specific Mouse Models for Pineoblastoma

In the past several years, genetically engineered mouse models (GEMM) for RB- [94], DICER1- [94] and DROSHA- [95] PBs with targeted deletion of the corresponding murine genes together with p53 in the pineal gland, have been reported. A model for MYC-PB is yet to be developed. In addition, PB progression and genomic alterations were characterized in cyclin D1-driven PB.

### 9.1. RB1-Deficient Pineoblastoma

Tryptophan hydroxylase (TPH)-SV40 T antigen fusion transgenic mice develop pineal tumors at 12–15 weeks of age [96]. SV40 large Tag binds and sequesters RB protein family (pRB, p107, p30), p53 and other factors and therefore does not mimic the human disease. Approximately 40% of Rb^+/−^:p53^−/−^ mice with an heterozygous knockout allele of *Rb* and homozygous null mutations in p53 develop PB [97]. These mice also develop other lesions, including pituitary tumors (94%), thyroid tumors (60%), lymphomas (39%), sarcomas (43%), islet cell tumors (23%), bronchial hyperplasia (38%) and retinal dysplasia (41%) [97]. The concurrent development of multiple tumors complicates and limits the analysis of PB in this mouse model. To target the pineal gland, an Interphotoreceptor Retinol Binding Protein (IRBP) promoter was used to drive the expression of the Cre recombinase to rod and cone photoreceptor cells as well as pinealocytes. IRBP-Cre:Rb^flox/flox^:p53^+/−^ and IRBP-Cre:Rb^flox/flox^:p53^−/−^ mice developed multiple brain lesions including anterior lobe tumors (63% in IRBP-Cre:Rb^flox/flox^:p53^+/−^ mice; 15% in IRBP-Cre:Rb^flox/flox^:p53^−/−^ mice), melanotrope tumors (2%; 0%), pituitary tumors (8%; 0%), PB (15%; 62%), and other unclassified lesions (11%; 85%) [98]. No mouse model that develops PB with high specificity and penetrance has been available till recently.

Our group has made the fortuitous discovery that the Whey Acidic Protein, WAP-Cre deleter line [99], commonly used to target mammary lobuloalveolar progenitors during the estrous cycle and pregnancy, is expressed in the pineal gland in both male and female mice. WAP-Cre:Rb^flox/flox^:p53^flox/flox^ mice develop large PBs that protrude the head and disseminated to the spinal cord with a short latency (median of 133 days) and 100% penetrance [94] (Figure 2). No tumors were developed when *Rb* or *p53* alone, or both *Pten* and *p53* were deleted, indicating specific susceptibility to *Rb* plus *p53* deletion for PB development. Histologically, the *Rb*-deficient PBs were highly proliferative, displaying limited cytoplasm, pleomorphic nuclei and rosette-like features. Early lesions were detected in the pineal gland in 18-day-old mice (earliest stage analyzed) using histology and mT/mG reporter mice. Male and non-pregnant female Wap-Cre:Rb^flox/flox^:p53^flox/flox^ mice develop PB at a similar frequency, indicating Wap-Cre transgene expression in the pineal gland is not hormonally regulated as in the mammary gland.

Progression of PB was reported in one case to concur with high expression of TP53 immuno-staining, indicative of stabilizing p53 mutation [100]. Consistent with this, WAP-Cre:Rb^flox/flox^:p53^lsl_R270H/flox^ mice with Rb-loss plus p53^R270H^ mutation in the DNA-binding domain showed a similar latency but increased rates of metastasis to leptomeningeal surfaces of brain and spinal cord [94]. Occasionally, these mice developed both PB and pituitary tumors. We have recently found that after crossing with a mT/mG reporter strain, Wap-Cre:Rb^flox/flox^:p53^flox/flox^:mT/mG mice develop additional, yet to be classified, brain lesions with longer latency (unpublished). This may be due to the immunogenic effect of GFP that is induced in the pineal gland/PB in mT/mG mice [101,102] or the mixed background and genetic modifiers in these composite mice which may affect penetrance.

The rationale for generating *Rb*/*p53* double mutation to model PB is based on the aforementioned observation that progression to full-blown PB in a human patient was associated with high TP53 immuno-staining, suggesting a stabilizing mutation and that *p53* loss is a late event that drives metastatic dissemination [100]. Although frequent *p53* mutations have not been identified in primary PB, this tumor suppressor may be mutated in metastases or inactivated by other mechanisms (e.g., protein stability and miRNA regulation). In addition, *RB1* and *p53* are often lost together in aggressive forms of cancer including those of breast, ovary, pancreas and prostate [103,104,105], and nearly all *Rb*-deficient mouse models including brain tumor models require additional mutation in *p53* [106,107,108,109]. However, other *Rb*-deficient mouse models require mutations in one of its relatives; Rb-like (*Rbl*) 1 (*p107*) and *Rbl2* (*p130*), not in *p53*. Indeed, while *RB1* mutation alone induces retinoblastoma in children, *Rb* plus *p107* or *p130*, not *p53*, induces retinoblastoma in mice [110,111]. Importantly, *RBL2*/*p130* is often lost in both retinoblastoma and PB as part of chr 16q deletion [44,112]. Thus, it is possible that *Rb* plus *p130* mutation (on a wild-type p53 background) will induce PB that faithfully resemble the human disease.

To identify new vulnerabilities for metastatic 1-deficient PB, we employed mRNA expression data from the two mouse PB models (WAP-Cre:Rb^flox/flox^:p53^flox/flox^ and WAP-Cre:Rb^flox/flox^:p53^lsl_R270H/flox^) to calculate connectivity scores [113] via GSEA and Genome-Wide Connectivity (GWC) map as described [114,115]. In silico analysis consistently identified tricyclic antidepressants drugs, some of which are FDA-approved antidepressants or antipsychotics, as top inhibitors for both *Rb*/*p53*-deleted and *Rb*/*p53*-mutated PBs [94]. These included nortriptyline at the top of the list as well as promazine, norcyclobenzaprine and amitriptyline. Nortriptyline suppressed PB growth by inhibiting autophagy through disruption of the lysosome, leading to accumulation of non-functional autophagosome, cathepsin B release and cell death. Nortriptyline suppressed both mouse and human primary PB growth in culture. This tricyclic antidepressant also suppressed PB development in our preclinical models and further synergized with gemcitabine to extend overall survival. Notably, nortriptyline alone and even in combination with gemcitabine delayed but did not eradicate PB, albeit very early administration of the drugs at the onset of tumor development, as monitored by MRI. This may point to the preferential killing of non-PB cancer-initiating cells, allowing PB cancer-initiating cells to survive and grow, lack of sufficient local drug concentration to effectively kill tumor cells, or rapid selection and emergence of drug-resistant variants due to absence of p53.

Although both *RB1* loss and *TP53* loss are not directly druggable, we and others demonstrated that the ATM/ATR-CHK-WEE1-CDC25-AURORA A/B KINASE pathway is critical for survival of *RB1*/*TP53*-deficient TNBC, and that antagonists of this pathway, including inhibitors for CDC25, WEE1 (adavosertib; MK1775) and AURORA A KINASE (alisertib) efficiently kill diverse TNBC including *RB1*/*TP53*-deficient lesions [72,116,117]. *RB1* loss also induces the immune checkpoint modulators poliovirus receptor (PVR) and programmed death ligand 1 (PD-L1) [118], and metabolic reprograming including mitochondrial protein translation [106,119], all of which are amenable to therapeutic interventions. Furthermore, in triple-negative breast cancer, while *RB1* loss does not significantly alter sensitivity to multiple (over 3400) inhibitory compounds, it selectively increases sensitivity to gamma-irradiation, and moderately to doxorubicin and methotrexate compared to *RB1*-positive lines [120]. Whether such inhibitors or gamma-irradiation either alone or in combination with other drugs including autophagy inhibitors like nortriptyline can effectively halt progression of *RB1*-deficient PB is yet to be demonstrated.

### 9.2. DICER1-Deficient Pineoblastoma

Using WAP-Cre, we also deleted *Dicer1* in the pineal gland together with *p53*; WAP-Cre:Dicer1^flox/flox^:p53^flox/flox^ mice developed PB with partial penetrance (31.6%) and average latency of 270 days [94]. As in human PB, both alleles of *Dicer1* had to be deleted to obtain PB. Despite differences in latency, Dicer1-PB exhibited similar histology of pleomorphic nuclei, scarce cytoplasm and rosette structures as seen in Rb/p53-deficient PBs. In addition, despite their large size, the Dicer1 PBs did not protrude the brain as did the Rb lesions. Gene set enrichment analysis (GSEA)-based principal component analysis (PCA) revealed that *Dicer1*/*p53*- and *Rb*/*p53*-deficient PBs clustered closely with mouse retinoblastoma and human PB, but as expected, distant from glioblastoma [94,121]. As noted, the 5-year OS for PB-miRNA1 and PB-miRNA2 patients is approximately 70%, and 100%, respectively, suggesting that WAP-Cre:Dicer1^flox/flox^:p53^flox/flox^ mice faithfully model PB-miRNA1 (Figure 2).

Importantly as described above for *RB1*-deficient mouse PB, connectivity mapping analysis of WAP-Cre:Dicer1^flox/flox^:p53^flox/flox^ mice also identified nortriptyline as a leading putative therapeutic alongside other tricyclic/anti-psychotic drugs (chlorpromazine, clomipramine, fluphenazine and norcyclobenzaprine) [94]. These results point to autophagy as a major vulnerability in diverse PB subtypes. Thus, the kinetics, penetrance and histology of *Rb*- and *Dicer1*-deficient PB highly resemble the human disease, offering great preclinical models in immune-competent mice.

### 9.3. DROSHA-Deficient Pineoblastoma

An unpublished work submitted to bioRxiv describes the generation of IRBP-Cre:Rb^flox/flox^:p53^flox/flox^, IRBP-Cre:p53^flox/flox^:Dicer1^flox/flox^ and IRBP-Cre:p53^flox/flox^:Drosha^flox/flox^ mice [95]. Latency and penetrance was similar to those observed in WAP-Cre:Rb^flox/flox^:p53^flox/flox^ and WAP-Cre:Dicer1^flox/flox^:p53^flox/flox^ mice [94] (Figure 2). Thus, deletion of *Rb* and *Dicer1* via two different promoter (WAP-Cre and IRBP-Cre) induces PB with similar histology and penetrance that mimic the human disease.

The Dicer1 and Drosha-PB retain functional, phosphorylatable pRb and are accordingly sensitive to the CDK4/6 inhibitor palbociclib [95]. These tumors exhibited loss of diverse microRNAs including the let-7/miR-98-5p family, with de-repression of their target genes: Bach1, Tgfbr1 and the transcription factor Plagl2 (PLAG1 Like Zinc Finger 2). Indeed, the Plagl2 target gene, Igf2, was highly overexpressed, and the downstream RAS and PI3K pathways were induced in the Drosha PB. In accordance, growth of these tumors in subcutaneous xenograft assays was attenuated by ceritinib, an inhibitor of IGF1R (insulin-like growth factor 1 receptor) and INSR (insulin receptor). Notably, however, ceritinib is 50-fold more potent against ALK (anaplastic lymphoma kinase), suggesting that this drug may reduce Igf2-mediated downstream signaling to RAS and PI3K pathways by indirectly inhibiting ALK, which also activate these pathways [122]. Whether ceritinib and palbociclib can attenuate PB development in the native IRBP-Cre:p53^flox/flox^:Drosha^flox/flox^ mice or in orthotopic rather than subcutaneous models remains to be determined.

### 9.4. cMYC-Driven Pineoblastoma

So far, no GEMM for MYC- or FOXR2-driven PB has been reported (Figure 2). Such immune-competent GEMM, generated using WAP-cre or IRBP-Cre and available inducible transgenes [86,123,124,125,126], would be invaluable in investigating candidate therapeutic regimens such as those developed for other MYC-driven brain tumors. Thus, PI3K/mTOR inhibitors like NVP-BEZ235 were shown to inhibit tumor cell proliferation, and further sensitize glioma stem cells to radiotherapy through the activation of autophagy, increased apoptosis, cell cycle arrest and reduced DNA repair [127]. A drug screen for MYC-driven medulloblastoma (MB) identified histone deacetylase inhibitors (HDACi), which suppress growth in part by inducing expression of the *FOXO1* tumor suppressor gene, and further synergize with PI3K inhibitors to suppress MB in vivo [128]. Indeed, the HDAC inhibitor panobinostat reduces tumor cell proliferation and induces apoptosis in high-risk neuroblastoma [129] and medulloblastoma [130]. BET bromodomain inhibition via JQ1 and related inhibitors show promising effects in preclinical models of MYC-driven neuroblastoma, glioblastoma and medulloblastoma [131,132,133].

MYC amplification promotes cancer by inducing transcription via enhancers and super-enhancers [134,135], which in turn produces topological constraints that interfere with efficient transcription. This potential hindrance can be resolved by topoisomerase 1 and RAD1 via the DNA double-strand breaks (DSBs) machinery [136]. Consistent with this, inhibition of topoisomerase 1 leads to abnormal R-loop accumulation and synthetic lethality with MYC-driven cancer [71]. As noted, several clinical trials on brain cancer including PBs such as NCT02596828 (Table 1) test for the effect of topo 1 inhibitors like irinotecan. Re-analysis of the results for the subset of MYC-driven tumors in these trials may reveal a subtype-specific vulnerability.

Although MYC alone is not directly druggable, several small molecule inhibitors that block the heterodimerization of cMYC, a basic-helix-loop-helix-leucine zipper (bHLH-LZip) transcription factor, with its partner MAX [137]. Several such MYC-MAX inhibitors including KJ-Pyr-9 [138], MYCMI-6 [139] and MYCi975 [140,141] effectively suppress MYC-mediated transcription and MYC-driven tumors. In addition, it has been shown that in breast cancer driven by *cMYC* amplification, combination of MYC plus mTOR inhibitors (AZD8055) synergized to suppress tumor growth in vivo [142]. Such targeted therapy may be combined with inhibitors against MYC-induced metabolic [143] and immune [144,145] reprograming. A mouse model for MYC-driven PB would enable testing for such candidate inhibitors as well as perform non-biased screens on derived primary MYC-PB cells.

### 9.5. Cyclin D1-Driven Pineoblastoma

The tumor suppressor pRB is inactivated during the cell cycle through sequential phosphorylation by cyclin D and Cyclin E-associated kinases (CDK4/6 and CDK2, respectively) [119,146]. In breast cancer and other tumor types, pRB is lost either by direct mutations/deletions/promoter methylation and silencing of the *RB* gene, or by hyper-phosphorylation of the pRB protein through loss of *p16^INK4A^* or amplification and activation of cyclin D1, cyclin E and CDK4 [59,106]. In PB, there is no evidence for pRB inactivation through hyper-phosphorylation of the protein. Instead, the *RB* gene is directly lost by mutations/deletions. Nonetheless, Irbp-cyclin D1 transgenic mice exhibit pRb hyper-phosphorylation and hyperproliferation that is accompanied by cell senescence; combined cyclin D1 overexpression plus p53 null mutation leads to continuous growth and development of PB [100].

The Irbp-cyclin D1:p53^−/−^ mice have been further interrogated to define the temporal progression of PB and relationship between chromatin accessibility and expression profiles [147]. Interestingly, despite rapid tumor formation, there is a steady increase in copy number variations (CNVs) including amplifications and deletions from p10 to p49 and p90, the time points at which PBs were analyzed. These include an amplification of *Hs1bp3*, a negative regulator of autophagy [148], an observation that is in line with the abovementioned evidence for aberrant autophagy in *Rb*- and *Dicer1*-deficient PB [94], and may point to a general defect in autophagy in PB.

Together, these murine models closely resemble human PB at the molecular level, histology, penetrance, and tumor progression, providing powerful tools for further dissecting mechanisms of tumorigenesis, identify specific vulnerabilities and testing novel therapeutic strategies.

## 10. Pineoblastoma Cell of Origin

The cell of origin of cancers may be stem cells, progenitors or differentiated cells that are induced to de-differentiation due to oncogenic insults. In medulloblastoma (MB), the four major subtypes—SHH, WNT, group 3 and group 4—arise from different cells of origin and developed through distinct oncogenic pathways. Thus, WNT MBs originate from mossy fiber neuron (MFNs) in the dorsal brainstem; SHH MBs from granule neuron progenitors (GNPs) in the external granular layer (EGL); group 3 from stem cells and nascent unipolar brush cells (UBCs) in the rhombic lip ventricular zone (RLVZ) and rhombic lip subventricular zone (RLSVZ), respectively; whereas group 4 from nascent UBCs in RLSVZ (reviewed in [149]).

The identification of a tumor cell of origin may guide novel therapeutic approaches. For example, group 3 MB, which is associated with exceedingly poor survival, originates from transformation of Protogenin-positive (PRTG^+ve^), MYC^high^, NESTIN^low^ stem cells in the human embryonic hindbrain that subsequently localizes to the ventricular zone of the rhombic lip [150]. Ablation of PRTG^+ve^ cells or anti-PRTG CAR T cell therapy attenuated group 3 MB growth in xenograft models [150].

In contrast to the diverse cellular origin of MB, all PB subtypes appear to originate from pinealocytes. Indeed, both Wap-Cre- and IRBP-Cre-mediated deletions of *Rb*, *Dicer1* and *Drosha* induce histologically similar PB, suggesting a common cell of origin. In preliminary results, we found that early *Rb*/*p53*-deleted PB lesions stained positive for 5-hydroxytryptamine receptor (5-HT, serotonin receptor), which marks matured pinealocytes, but negative for pax6 (pinealocyte precursor cells), nestin (neuronal stem cells) and the microglia marker OX-42 [94]. The IRBP-Cre transgene is also thought to target pinealocytes. More rigorous analysis is required to establish this lineage as the susceptible cell for PB development in mice and humans, and whether depletion or targeting these cells in combination with subtype-specific therapies can attenuate PB tumor growth as described above for group 3 MB.

Finally, in retinoblastoma, the loss of *RB1* alone appears to induce benign, hypoproliferative retinoma, but additional oncogenic insults are required to transform these cells into highly aggressive tumors. Retinoma exhibit high expression of the CDK inhibitor p16^INK4B^ and the pRB relative, p130, both of which are associated with induction of cell senescence. Additional alterations including reduction in p16^INK4B^ and p130 levels, loss of the tumor suppressor genes CDH11 and p75NTR, and copy number gain and high expression of the oncogenes MYCN, E2F3, DEK, KIF14 and MDM4, can induce clonal expansion and full-blown retinoblastoma [151]. Such cooperating oncogenic alterations serve as candidate therapeutic targets for combination therapies. Pineocytoma is a benign, slowly growing tumor of the pineal gland identified mostly in adults [21]. Whether pineocytoma is a precursor for PB as retinoma is for retinoblastoma and what additional genomic and epigenomic alterations are associated with progression of each PB subtype in addition to the main oncogenic drivers are yet to be defined.

## 11. Future Directions

A major challenge in understanding the biology of PB is the scarcity of the disease and limited availability of high-quality tumor tissue. Advances in DNA and RNA sequencing from minute and degraded samples, including paraffin sections, will enable robust analysis, particularly, mutations, epigenomic and proteomic alterations that may cooperate with the major drivers of the disease.

Despite these limitations, two significant breakthroughs in PB research have emerged in the past five years. First, molecular stratification revealed that PBs can be divided into four major groups driven by loss of microRNA regulators DICER1, DROSHA or DGCR8, loss of the tumor suppressor *RB1* or somatic amplification/stabilization of the cMYC oncogene [43,44,45]. While miRNA-PBs exhibit relatively good outcomes, the survival and quality of life of infants and young children with RB1- and MYC-PBs remain dismal, highlighting the need for better diagnostic and therapeutic modalities. Importantly, as discussed herein, results from completed, ongoing and future clinical trials should be re-evaluated for possible effects on specific PB subtypes.

Second, subtype-specific GEMMs for each PB subtype, except MYC-PBs, have been developed, and will drive forward precision medicine. Two different studies in three models (RB, Dicer1, cyclin D1) identified autophagy as potential liability that can be exploited therapeutically [94,95,147]. These immune-competent mouse models will be instrumental in preclinical trials for drug combinations that can target specific vulnerabilities in each PB subtypes including the major drivers and cooperating oncogenic events to successfully treat this devastating pediatric brain cancer.

## Figures and Tables

**Figure 1 cancers-17-00720-f001:**
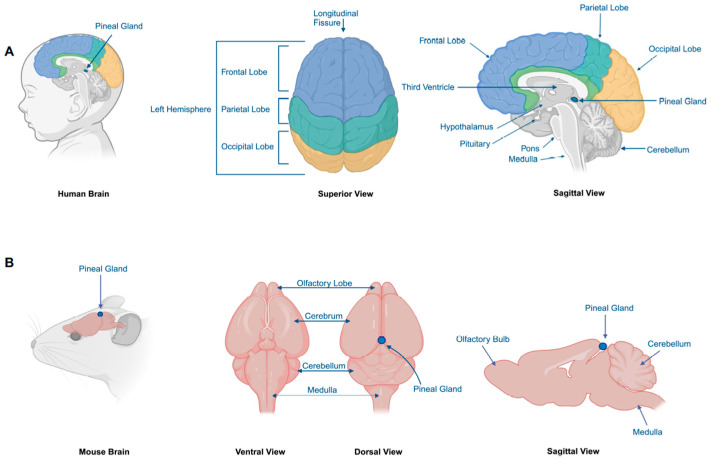
Distinct locations of the pineal gland in the human (**A**) and mouse (**B**) brains.

**Figure 2 cancers-17-00720-f002:**
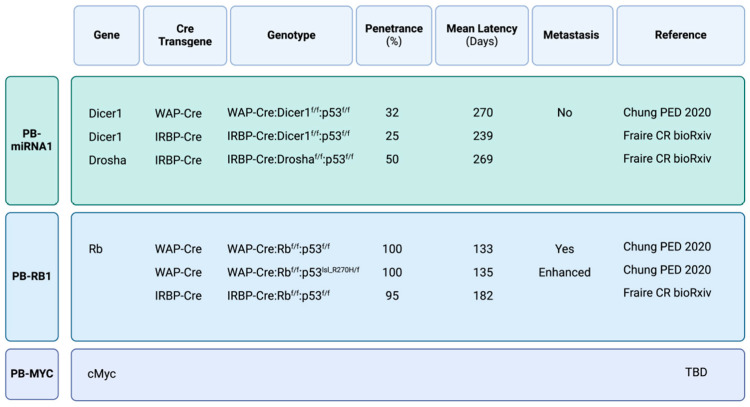
Mouse models for the major molecular subtypes of pineoblastoma [94,95]. See text for details. A mouse model for MYC-PB is yet to be developed (TBD).

**Table 1 cancers-17-00720-t001:** Ongoing clinical trials for pineoblastoma.

NCT Number	Study Status	Interventions	Sponsor	Age
**NCT01063114**	ACTIVE_NOT_RECRUITING	RAD: proton beam radiation	Massachusetts General Hospital	CHILD, ADULT
**NCT00867178**	COMPLETED	RAD: 3-D conformal radiation Therapy vorinostat and isotretinoin; vincristine; cisplatin; cyclophosphamide; etoposide	National Cancer Institute (NCI)	CHILD
**NCT06193759**	RECRUITING	BIO: Multi-tumor antigen specific cytotoxic T lymphocytes (TSA-T) against personalized tumor-specific antigens (TSA)	Children’s National Research Institute	CHILD
**NCT03638167**	ACTIVE_NOT_RECRUITING	BIO: EGFR806-specific chimeric antigen receptor (CAR) T cell	Seattle Children’s Hospital	CHILD, ADULT
**NCT03500991**	ACTIVE_NOT_RECRUITING	BIO: HER2-specific chimeric antigen receptor (CAR) T cell	Seattle Children’s Hospital	CHILD, ADULT
**NCT04185038**	RECRUITING	BIOL: SCRI-CARB7H3(s); B7H3-specific chimeric antigen receptor (CAR) T cell	Seattle Children’s Hospital	CHILD, ADULT
**NCT03382158**	RECRUITING	International PPB/DICER1 Registry	Children’s Hospitals & Clinics of Minnesota	CHILD, ADULT
**NCT05934630**	ACTIVE_NOT_RECRUITING	Testing Cerebrospinal Fluid for Cell-free Tumor DNA in Children, Adolescents, and Young Adults With Brain Tumors	Pediatric Brain Tumor Consortium	CHILD, ADULT
**NCT05064306**	AVAILABLE	DRUG: 131I-omburtamab	Memorial Sloan Kettering Cancer Center	CHILD, ADULT
**NCT00602667**	ACTIVE_NOT_RECRUITING	DRUG: Induction Chemotherapy: Low-Risk Therapy: High-Risk Therapy: Intermediate-Risk Therapy	St. Jude Children’s Research Hospital	CHILD
**NCT06357377**	NOT_YET_RECRUITING	DRUG: NEO100	Neonc Technologies, Inc.	CHILD, ADULT
**NCT02574728**	RECRUITING	DRUG: Sirolimus: Celecoxib: Etoposide: Cyclophosphamide	Emory University	CHILD, ADULT

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
