# Peer review of "Recent Advances in Pineoblastoma Research: Molecular Classification, Modelling and Targetable Vulnerabilities"

_cancers, 2025, doi:10.3390/cancers17050720_

Round 1

Reviewer 1 Report

Comments and Suggestions for Authors

The authors present a comprehensive and well written summary of the current knowledge of the clinical and molecular characteristics of pineoblastoma, including updated information regarding molecular subgroups and genetically accurate mouse models for pre-clinical study. They have done a fine job synthesizing the current literature and knowledge on the subject, as well as providing their own data and experience in modeling this disease. As a result, this paper is a useful update on this difficult to treat pathology.

Author Response

The authors present a comprehensive and well written summary of the current knowledge of the clinical and molecular characteristics of pineoblastoma, including updated information regarding molecular subgroups and genetically accurate mouse models for pre-clinical study. They have done a fine job synthesizing the current literature and knowledge on the subject, as well as providing their own data and experience in modeling this disease. As a result, this paper is a useful update on this difficult to treat pathology.

Response: We thank the Reviewer for the kind words.

Reviewer 2 Report

Comments and Suggestions for Authors

This is a succint review of Pineoblastoma with special regard on molecular classification, modelling and targetable vulnerabilities.

Besides the fact that the new WHO Classification of CNS Tumors should be cited, I did not find any point which needs further clarification.

Author Response

This is a succinct review of Pineoblastoma with special regard on molecular classification, modelling and targetable vulnerabilities. Besides the fact that the new WHO Classification of CNS Tumors should be cited, I did not find any point which needs further clarification.

Response: We thank the Reviewer for the kind words and for suggesting adding a citation to WHO Classification of CNS Tumors. We added the following reference:

Gaillard F, Deng F, Gagen R, et al. WHO classification of CNS tumors. https://doi.org/10.53347/rID-2277

Reviewer 3 Report

Comments and Suggestions for Authors

The review covers several modern topics related to pineoblastomas. First, the summary of current trials is informative and useful. Second, the part related to the different models of pineoblastomas is detailed. Furthermore, the cell-of-origin paragraph raises essential questions. However, the review lacks the explicated general idea and red line throughout the manuscript. It is not clear what conclusion can be drawn from the text or what the reason is to arrange all these topics together in one article. I advise authors to provide reasoning for simultaneous consideration of all of the parts, highlight some questions in the abstract and introduction to help readers identify the information that can be found here, and state clear conclusions and future directions from all the summarized material.

A couple of additional, minor, issues:

A lot of too-general phrases detected that could be eliminated without loss of information to make the article concise:

1) Chapter 2 looks too general for this article and seems to be just a filler. Consider constriction, (maybe)removing the illustration, and merging with the introduction.

2) Introductory phrases in Chapter 5 (general words on metastasis). The second paragraph here also seems to be redundant. Thus, the chapter can be shortened and merged with the previous one.

Lines 94-96. Please check the text here, both the formatting and the style.

Author Response

The review covers several modern topics related to pineoblastomas. First, the summary of current trials is informative and useful. Second, the part related to the different models of pineoblastomas is detailed. Furthermore, the cell-of-origin paragraph raises essential questions. However, the review lacks the explicated general idea and red line throughout the manuscript. It is not clear what conclusion can be drawn from the text or what the reason is to arrange all these topics together in one article. I advise authors to provide reasoning for simultaneous consideration of all of the parts, highlight some questions in the abstract and introduction to help readers identify the information that can be found here, and state clear conclusions and future directions from all the summarized material. A couple of additional, minor, issues: A lot of too-general phrases detected that could be eliminated without loss of information to make the article concise: 1) Chapter 2 looks too general for this article and seems to be just a filler. Consider constriction, (maybe)removing the illustration, and merging with the introduction. 2) Introductory phrases in Chapter 5 (general words on metastasis). The second paragraph here also seems to be redundant. Thus, the chapter can be shortened and merged with the previous one. Lines 94-96. Please check the text here, both the formatting and the style.  

Response: We thank the Reviewer for the thoughtful review.

a. With regards to “explicated general idea and red line” – we stressed the take-home-message in the Abstract and throughout the manuscript that “Pineoblastoma, which has so far been treated as a single disease,  can be segregated by multi-omic analysis into 4 major subtypes and that mouse models for each of these subtypes has now been or is in the process of being generated”. This is a major progress that will usher a new era in translational research and new treatments for  pineoblastoma.   

b. Regarding the various topics covered by the review – we see this as a strength as it consolidates recent basic and clinical progress in pineoblastoma research which may uncover new therapeutic modalities. For example, chapter 10 on the cell-of-origin of pineoblastoma may uncover new therapeutic avenue, as described in the first 2 paragraphs on group 3 medulloblastoma in this chapter. HOWEVER – we do agree with the reviewer that cell-of origin (Chapter 10) and clinical trial (chapter 6. Current treatment of pineoblastoma and ongoing clinical trials) are not mentioned in the Abstract. We have corrected this in the revised manuscript.  

c. “State clear conclusions and future directions” – this is captured in the SUMMARY paragraph which we now added – as well as throughout the manuscript and “Future directions”.  

Minor issues:

1. Chapter 2. We think that understanding the biology of the pineal gland (indeed any tissue and its respective cancer), is pertinent to pineoblastoma.  It also connects well with our discussion of the cell-of-origin of pineoblastoma.  The distinct location of the pineal gland in mouse and human is unique and relevant both to basic and translational scientists.  

2. Chapter 5 – metastasis. Again, we think that this discussion on metastasis is highly important for pineoblastoma. Although, as indicated, this cancer is detrimental even at the primary site, metastatic dissemination is lethal. Understanding mechanisms of dissemination such as the hematogenous route seen in MB, or TLR7-mediated interaction with sensory neurons observed in breast cancer may uncover novel therapeutic vulnerabilities.   

3. Lines 94-96. – Thanks – we have fixed that. 

Round 2

Reviewer 3 Report

Comments and Suggestions for Authors

There are no additional commentaries. The Summary helps to systematise the manuscript as it creates a clear picture of what will be mentioned and discussed.

Author Response

Thank you for valuable comments